# Doping-induced structural phase transition in cobalt diselenide enables enhanced hydrogen evolution catalysis

Ya-Rong Zheng[1], Ping Wu[2], Min-Rui Gao [1], Xiao-Long Zhang[1], Fei-Yue Gao[1], Huan-Xin Ju[3]
Rui Wu[1], Qiang Gao[1], Rui You[4], Wei-Xin Huang [4], Shou-Jie Liu[5], Shan-Wei Hu[3]
Junfa Zhu [3], Zhenyu Li [2] & Shu-Hong Yu [1]

Transition metal dichalcogenide materials have been explored extensively as catalysts to negotiate the hydrogen evolution reaction, but they often run at a large excess thermodynamic cost. Although activating strategies, such as defects and composition engineering, have led to remarkable activity gains, there remains the requirement for better performance that aims for real device applications. We report here a phosphorus-doping-induced phase transition from cubic to orthorhombic phases in $CoSe_2$. It has been found that the achieved orthorhombic $CoSe_2$ with appropriate phosphorus dopant (8 wt%) needs the lowest overpotential of 104 mV at 10 mA cm$^{-2}$ in 1 M KOH, with onset potential as small as −31 mV. This catalyst demonstrates negligible activity decay after 20 h of operation. The striking catalysis performance can be attributed to the favorable electronic structure and local coordination environment created by this doping-induced structural phase transition strategy.

[1] Division of Nanomaterials & Chemistry, Hefei National Research Center for Physical Sciences at the Microscale, CAS Center for Excellence in Nanoscience, Hefei Science Center of CAS, Collaborative Innovation Center of Suzhou Nano Science and Technology, Department of Chemistry, University of Science and Technology of China, Hefei 230026, China. [2] Hefei National Research Center for Physical Sciences at the Microscale, Synergetic Innovation Center of Quantum Information and Quantum Physics, University of Science and Technology of China, Hefei 230026, China. [3] National Synchrotron Radiation Laboratory, University of Science and Technology of China, Hefei 230029, China. [4] Hefei National Research Center for Physical Sciences at the Microscale, CAS Key Laboratory for Energy Conversion and Department of Chemical Physics, University of Science and Technology of China, Hefei 230029, China. [5] College of Chemistry and Materials Science, Anhui Normal University, Wuhu 241000, China. Correspondence and requests for materials should be addressed to M.-R.G. (email: mgao@ustc.edu.cn) or to S.-H.Y. (email: shyu@ustc.edu.cn)

Water electrolysis ($2H_2O \rightarrow 2H_2 + O_2$) powered by electricity from renewable sources holds great promise for sustainable hydrogen production that drives clean-energy devices such as fuel cells[1–3]. A considerable challenge toward large-scale utilization of this technology is the development of efficient and long-lasting catalysts able to accelerate cathodic hydrogen evolution reaction (HER)[2]. Although ultra-fast HER kinetics on platinum and its alloys are known, their adoption in scalable systems is plagued by high cost and low geological abundance. Recent advances in the search of new HER catalysts have shown that a range of transition metal dichalcogenides (TMD), such as $MoS_2$[4–9], $MoSe_2$[10], $WS_2$[11], $TaS_2$[12], $FeS_2$[13], $CoS_2$[14], $CoSe_2$[15–17], and $CoTe_2$[18] are attractive alternatives. Of these TMD catalysts, the intrinsic active site of $MoS_2$ was well identified[19], which stimulated strategies to tailor $MoS_2$ for promoted HER electrocatalysis through modulating material parameters such as defects[5], van der Waals interactions[6], morphology[7], composition[8], and crystal phases[9]. But although marked progress on $MoS_2$, how specific material parameter affects the activity of other TMD catalysts, for example, $CoSe_2$, remains poorly known.

As with $MoS_2$ and $WS_2$, structural phase transition from the 2H (trigonal prismatic) to 1T (octahedral) phase can be induced through chemical exfoliation of their layered compounds[11,20], which leads to enhanced HER activities owing to the strained metallic 1T phase. Another layered TMD material, $CoSe_2$, has recently been investigated as promising catalyst for $H_2$ production from water[15,16,21]. It is thought that the similar surface sites to active metal centers of hydrogenase[22], and the unsaturated coordination environment might account for the good HER energetics[6]. The crystal form commonly observed for $CoSe_2$ is the stable cubic phase ($c$-$CoSe_2$), which belongs to pyrite-type minerals with characteristic $Se_2^{2-}$ dumbbells, whereas $Co^{2+}$ occurs in octahedral coordination[23]. Intriguingly, experimental efforts have demonstrated that another $CoSe_2$ phase, namely orthorhombic marcasite ($o$-$CoSe_2$), can also be active for HER[17,24]. Although the cubic-to-orthorhombic phase change in $CoSe_2$ can be achieved in principle through rotating half of the $Se_2^{2-}$ groups[25], methods capable of realizing this transition are rare, which limits opportunities of switching its properties for energy applications, such as electrocatalysis.

In this work, we show the experimental observation of a phosphorus-doping-induced phase transition from $c$-$CoSe_2$ to $o$-$CoSe_2$ with controllable P-doping levels (denoted as $o$-$CoSe_2$|P). The choice of phosphorus as dopant is based on its weaker electronegativity in comparison to selenium that might tune the $d$ electron number on Co cations, which was thought to influence the structural phases between pyrites and marcasites[25,26]. We find that the resultant $o$-$CoSe_2$|P (8 wt%) is one of the most efficient low-cost materials for catalyzing the HER in alkaline electrolyte, in which the HER kinetics are two orders of magnitude slower than that in acid on platinum[1]. Experimental and theoretical studies illustrate that the developed $o$-$CoSe_2$|P catalyst offers an optimal electronic structure and local coordination environment after phase transition, leading to the substantial energetic benefit for HER. Our findings raise the possibility in accessing advanced electrocatalysts through element-doping-induced structural phase transition.

## Results

### Synthesis and characterization of $o$-$CoSe_2$|P. In our experiment, we achieved marcasite-type $o$-$CoSe_2$|P by annealing as-synthesized $c$-$CoSe_2$ nanobelts[27] (Supplementary Fig. 1) with $NaH_2PO_2 \cdot H_2O$ under an argon atmosphere, which was decomposed to $PH_3$ species in situ that enable the P-doping and phase transition (Fig. 1a and Methods). We note that similar pyrite-type $CoPS$[28] and Se-doped $NiP_2$[29] have been reported, but no phase change was observed. Scanning electron microscopy (SEM) and transmission electron microscopy (TEM) images of the obtained sample show decent belt-like morphology (Fig. 1b, c), inherited from the $c$-$CoSe_2$ precursors. Close-up inspection of this sample reveals porous structures (Fig. 1d and Supplementary Fig. 2) composed of long and narrow holes with sizes ranging from 1 to 15 nm (Supplementary Fig. 3). These holes were formed owing to the different diffusion rates of P and Se during transition process upon heating. Selected-area electron diffraction (SAED) patterns (inset in Fig. 1c and Supplementary Fig. 4) show stretched, single-crystal-like diffraction spots that are readily distinguishable from the original pattern of $c$-$CoSe_2$, resulting from the porous orthorhombic structures. A high-angle annular dark field scanning TEM (STEM) image demonstrates the high crystallinity of $o$-$CoSe_2$|P with resolved lattice fringe of (130) planes, in which defects from hole edges can be observed (red arrow, Fig. 1e and Supplementary Fig. 5). In Fig. 1f we present X-ray diffraction (XRD) pattern that shows diffraction peaks clearly identical to that of marcasite $CoSe_2$ with orthorhombic phase (JCPDS 53–0449). By comparison with as-made pure orthorhombic $CoSe_2$, the shift of the (111) and (120) planes to relatively higher diffraction angles is due to the P replacing partial Se atoms, suggesting the formation of $o$-$CoSe_2$|P (Supplementary Fig. 6).

We further investigated the $o$-$CoSe_2$|P by STEM, energy-dispersive X-ray spectrum (EDX) elemental mapping, and electron energy-loss spectroscopy (EELS), which together evidence the presence of P that uniformly doped over the structure (Fig. 1g and Supplementary Fig. 7). The phase structure and doping level of the final product can be adjusted by careful control of the annealing temperature, time, and deposition amount of P during the heating process (Supplementary Figs. 8–11). As shown below, the optimal P content in $o$-$CoSe_2$|P for HER is ~8 wt%, on the basis of our inductively coupled plasma (ICP) atomic emission spectroscopy measurement.

### Doping-induced structural phase transition. Although orthorhombic-to-cubic phase change in $CoSe_2$ is widely affirmed[23,25,30], yet its reverse transition has not been reported. To observe the new transition process from pyrite-type $c$-$CoSe_2$ to marcasite-type $o$-$CoSe_2$|P, we monitored the temperature-dependent XRD patterns, in combination with other temperature-dependent characterizations, including X-ray photoelectron spectroscopy (XPS), EDX, and Raman spectroscopy (Fig. 2a–d). Figure 2a shows that when the annealing temperature reaches 300 °C, new XRD diffraction peaks (black arrows) that belong to $o$-$CoSe_2$ start to emerge from $c$-$CoSe_2$ matrix, indicating that the phase transition occurs. Complete phase transition of $c$-$CoSe_2$ to $o$-$CoSe_2$ with high crystallinity was observed at elevated temperature (400 °C) for a reaction of 30 min (Fig. 3a and Supplementary Fig. 12). No addition of P precursor in the reaction system, however, gives unchanged XRD pattern even at 400 °C (Fig. 3a, Supplementary Fig. 13), implying the critical role of P in inducing the phase transition. These results match well with P 2$s$ XPS and EDX data (Fig. 2b, c), taking into account that gradual increase in P content is responsible for yielding $o$-$CoSe_2$|P. Raman spectra were also recorded at selected temperature points (Fig. 2d), which show obvious Raman active peak at 189 cm$^{-1}$ for $c$-$CoSe_2$, corresponding to the Se–Se stretching mode[31]. This peak fades away with a new peak at 174 cm$^{-1}$ gradually dominant as the annealing temperature elevated from 300 to 400 °C, indicating the structural evolution from $c$-$CoSe_2$ to marcasite $o$-$CoSe_2$|P[32]. Raman spectrum also uncovers that if no P precursor was added

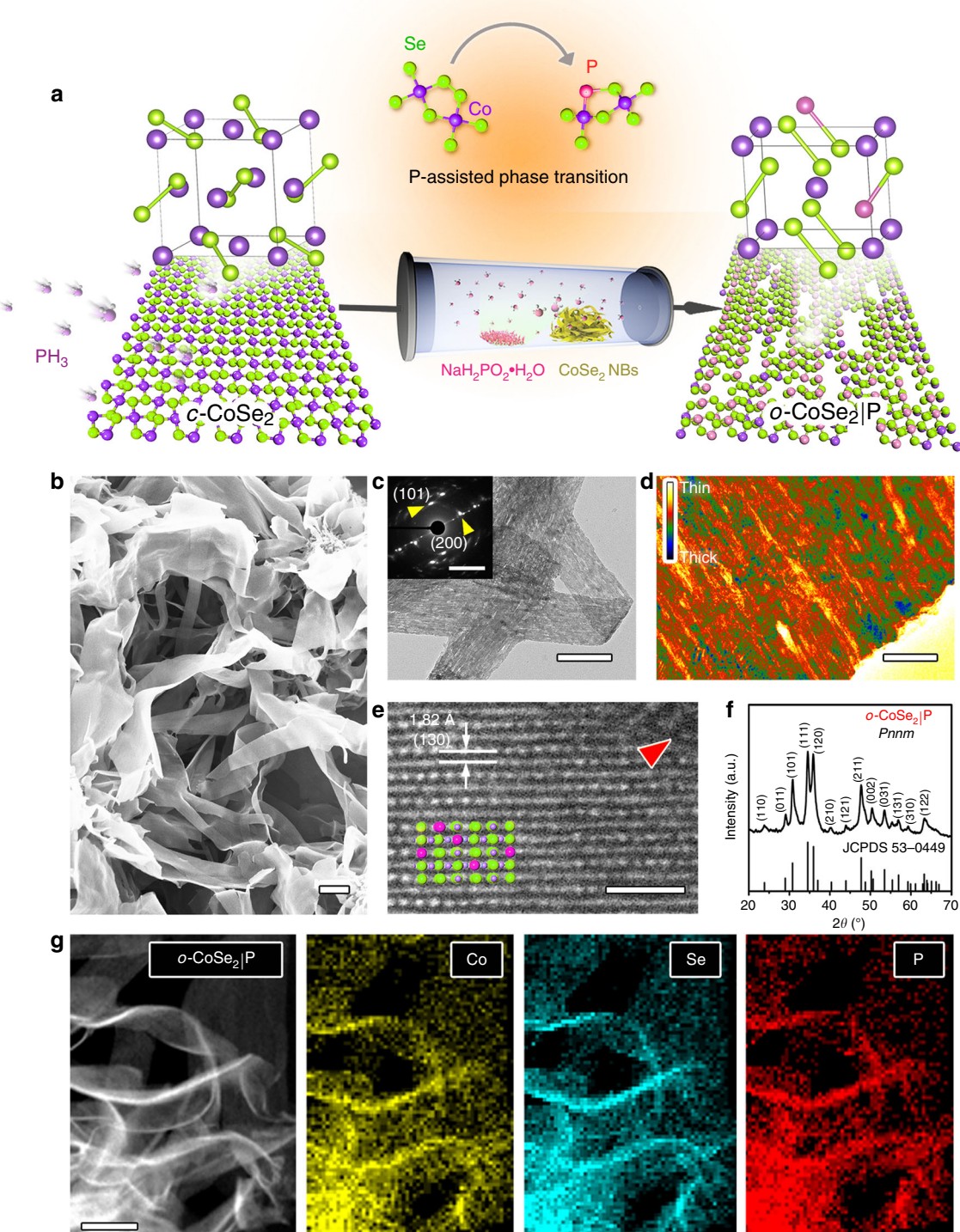

**Fig. 1** Synthesis and physical characterization of $o$-CoSe$_2$|P. **a** Schematic illustration of the P-doping-assisted structural phase transition from $c$-CoSe$_2$ to $o$-CoSe$_2$|P through an annealing process. Violet, green, and pink balls correspond to Co, Se, and P atoms, respectively. **b**, **c** SEM (scale bar, 400 nm) and TEM (scale bar, 200 nm) images of $o$-CoSe$_2$|P nanobelts. Inset in **c** shows corresponding SAED pattern. Scale bar, 5 nm$^{-1}$. **d** False-color HAADF image indicates the relative thickness of a porous $o$-CoSe$_2$|P nanobelt. Scale bar, 40 nm. **e** HAADF-STEM image of the $o$-CoSe$_2$|P. Red arrow in **e** indicates atomic defects at the pore edge. The $o$-CoSe$_2$|P crystal structure is shown in **e** with Co, Se, and P in blue, green, and pink, respectively. Scale bar, 1 nm. **f** Typical XRD pattern of the marcasite $o$-CoSe$_2$|P. **g** STEM-EDX elemental mapping of $o$-CoSe$_2$|P revealing clearly the homogeneous distribution of Co (yellow), Se (azure), and P (red), respectively. Scale bar, 200 nm

in the annealing process, $c$-CoSe$_2$ phase maintains even up to 400 °C, consistent with above observations (Fig. 2d).

Recent reports show that structural phase transition in TMD materials is commonly triggered by thermal[17] or chemical means[33]. A new electrostatic control over phase structure via electrostatic doping was lately demonstrated theoretically and experimentally in monolayer MoTe$_2$[34]. So far, phase transition from metastable $o$-CoSe$_2$ to $c$-CoSe$_2$ has been described through

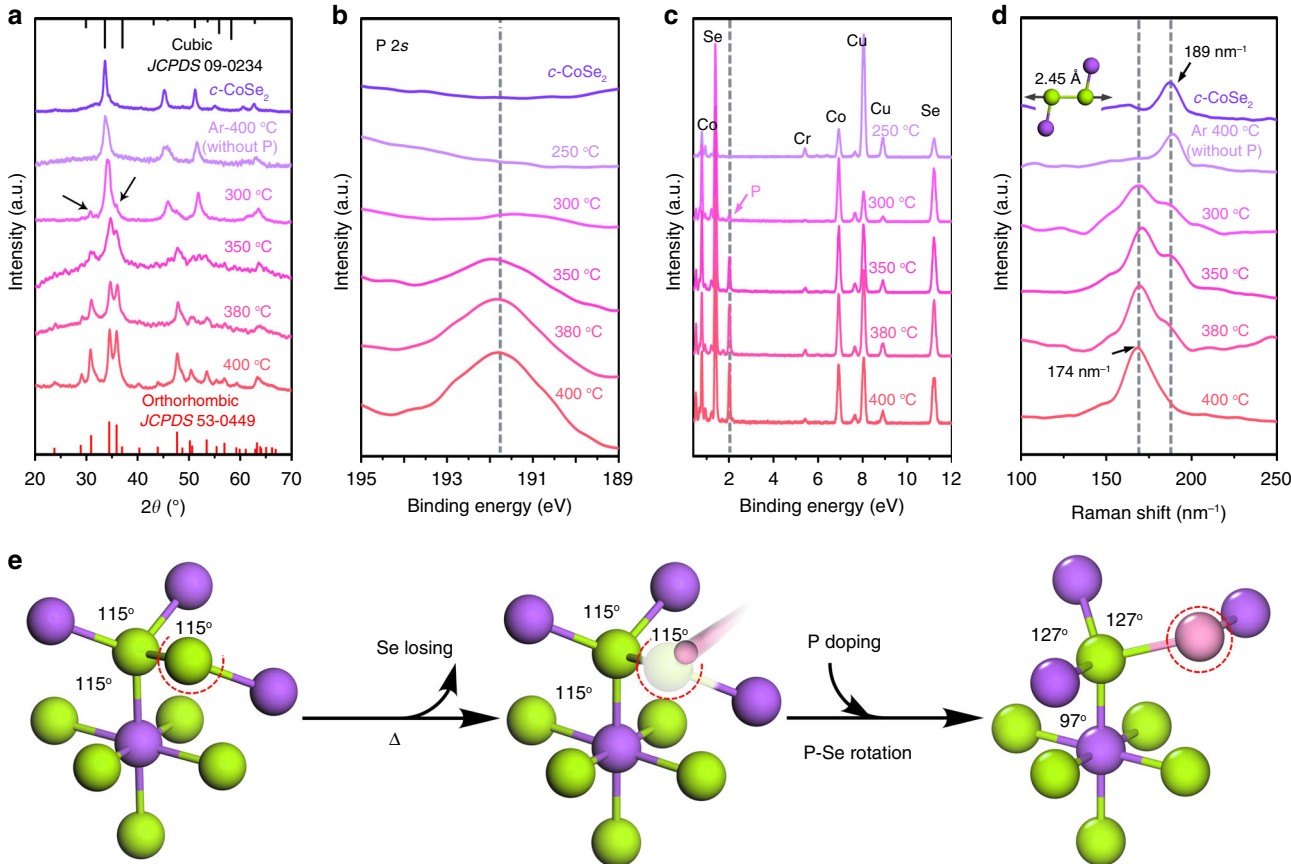

**Fig. 2** P-doping-assisted structural phase transition. **a–d** Temperature-dependent XRD patterns, P 2s XPS spectra, EDX spectra, and Raman spectra reveal the structural phase change from $c$-CoSe$_2$ to $o$-CoSe$_2$|P, respectively. **e** Schematic of the P-doping-assisted phase-transition process from cubic to orthorhombic phases in CoSe$_2$

thermal synthesis at 450 °C[17]; whereas, attempts to establish reverse conversion are rare, owing to the more thermodynamically stable $c$-CoSe$_2$ phase. Early studies showed that defects such as stacking faults in cubic pyrites might provide sites to drive energetically unfavorable marcasites[25]. In our study, annealing $c$-CoSe$_2$ at 400 °C results in the loss of some Se, leaving vacancy defects (Fig. 2e and Supplementary Fig. 14). These vacancies were quickly occupied by P generated in situ. Because of the weaker electronegativity of P in comparison to Se, its participation induces the formation of closer Co–P interactions and longer Se–Se(P) pairs. Such new bond reconstructions allow for tuned electrons in $d$ orbitals on Co, which was regarded as driving force to the rotation of Se–Se(P) pairs[26], thus giving rise to $o$-CoSe$_2$|P (Fig. 2e). Our P 2p and valence band edge XPS spectra clearly uncover the presence of Co–P and Se–P bonds (Fig. 4a, b, discuss later), confirming the role of P that aids the structural phase transition from stable $c$-CoSe$_2$ to metastable $o$-CoSe$_2$|P.

**Electrocatalytic HER activity and stability.** The TMD catalysts reported to date all investigate, with few exceptions[17,35], HER behaviors in acidic environments. We explore here the capability of HER electrocatalysis using $o$-CoSe$_2$|P in alkali, in which the HER kinetics are two orders of magnitude slower than that in acid on platinum. To this end, we examined the HER activity of $o$-CoSe$_2$|P on inert glassy carbon electrode in Ar-saturated 1 M KOH (pH 14) at ambient temperature; with reference measurements of other studied catalysts for comparison (see Methods). Polarization curves in Fig. 3a shows the onset potential (defined as the overpotential at 1 mA cm$^{-2}$) for H$_2$ evolution at −31 mV for $o$-CoSe$_2$|P, whereas the onsets were shifted substantially

negative for $c$-CoSe$_2$, and annealed $c$-CoSe$_2$, indicating energetical merits of $o$-CoSe$_2$|P catalyst. Figure 3a also shows that although inferior HER activity at low applied potentials, $o$-CoSe$_2$|P can far exceed Pt/C benchmark at high overpotentials (>170 mV), presumably owing to its highly porous structure that enables better mass-transfer process. The $o$-CoSe$_2$|P catalyst exhibits a low overpotential of 104 mV at 10 mA cm$^{-2}$; by comparison, overpotentials were 330 mV for $c$-CoSe$_2$ and 248 mV for annealed $c$-CoSe$_2$. Exchange current density ($j_0$), the most inherent measure of HER activity, was 0.43 mA cm$^{-2}$ for $o$-CoSe$_2$|P in 1 M KOH (Supplementary Fig. 15). This high $j_0$ agrees well with the large H$_2$ formation turnover frequency of 14.95 s$^{-1}$ for $o$-CoSe$_2$|P at 200 mV overpotential (Supplementary Fig. 16 and Supplementary Note 1). Tafel analysis (Fig. 3b) offers a slope of 179, 155, 69, and 112 mV dec$^{-1}$ for $c$-CoSe$_2$, annealed $c$-CoSe$_2$, $o$-CoSe$_2$|P, and Pt/C catalyst, respectively. The Tafel slope of 112 mV dec$^{-1}$ measured for Pt/C matches well with previous reports (113 mV dec$^{-1}$)[36], and the lower value of 69 mV dec$^{-1}$ gained for $o$-CoSe$_2$|P suggests its HER superiority as compared to Pt/C and other documented HER single catalysts (Supplementary Table 1). The Tafel slope of 69 mV dec$^{-1}$ also hints at a Heyrovsky-Volmer pathway that likely takes effect on $o$-CoSe$_2$|P catalyst[13]. The rotating ring disk electrode measurements show clear H$_2$ oxidation currents occurred on Pt ring at 0.5 V versus RHE, confirming the selective H$_2$ production on the studied catalysts (Fig. 3c). The H$_2$ production was further analyzed by gas chromatography, which shows that the detected amount H$_2$ gas is consistent with the theoretical value, corresponding to a Faradaic efficiency of ~100% (Supplementary Fig. 17). We note that doping bare $o$-CoSe$_2$ with P (~7.75 wt%), however, is unable to achieve the

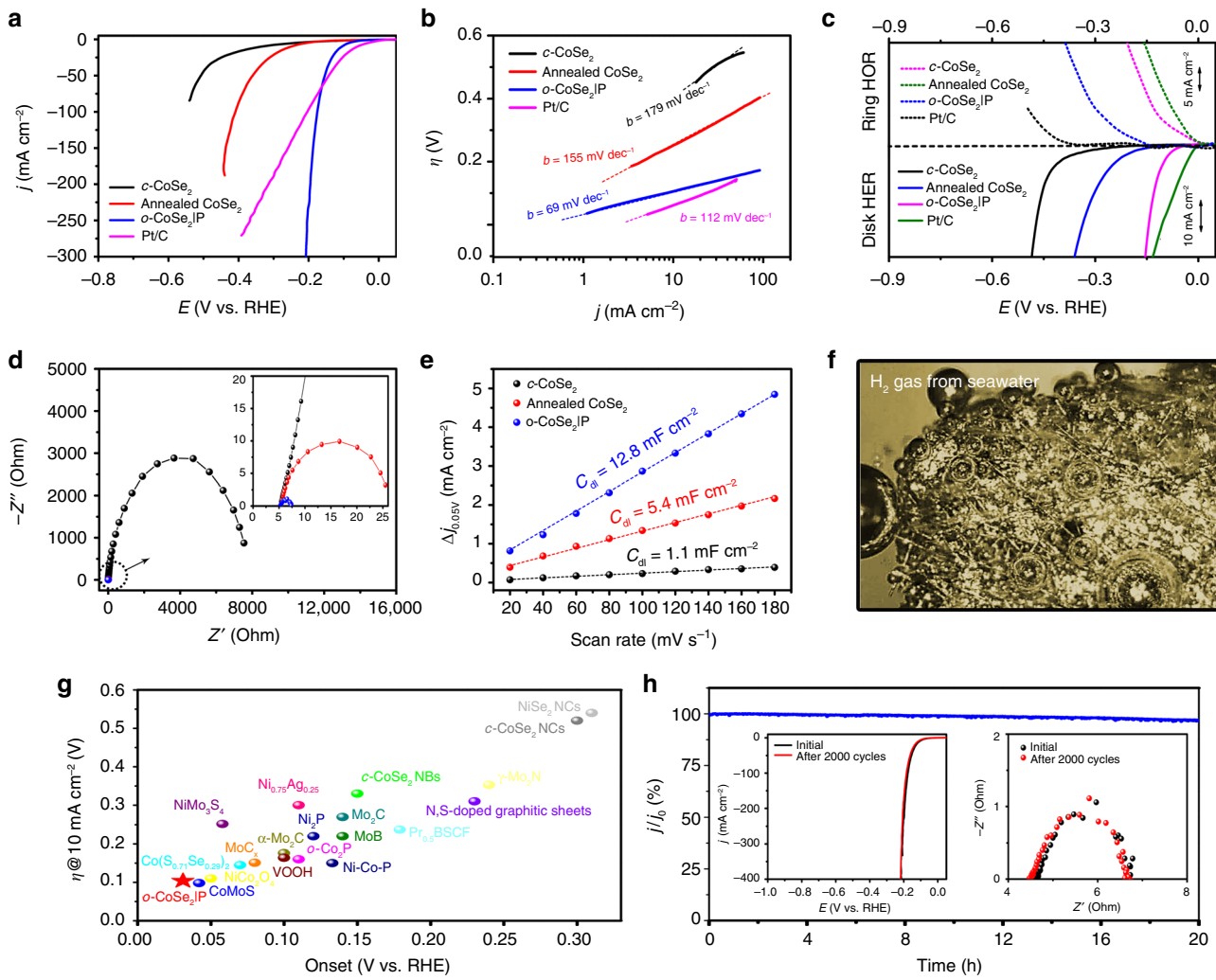

**Fig. 3** HER electrocatalysis on $o$-CoSe$_2$|P. **a** Polarization curves for the HER on $c$-CoSe$_2$, annealed $c$-CoSe$_2$, $o$-CoSe$_2$|P, and commercial Pt/C catalyst (20 wt %). Catalyst loading: ~1.02 mg cm$^{-2}$. Sweep rate: 10 mV s$^{-1}$. **b** Tafel plots for various catalysts derived from **a**. **c** Rotating ring disk electrode tests of H$_2$ evolution and oxidation on various catalyst-modified glassy carbon electrodes. The Pt-ring electrode was kept at 0.5 V for the oxidation of H$_2$ that was generated on the disk electrode. **d** EIS Nyquist plots of $c$-CoSe$_2$, annealed $c$-CoSe$_2$, and $o$-CoSe$_2$|P catalysts. Inset shows Nyquist plots at high-frequency range. $Z'$ is the real impedance and $Z''$ is the imaginary impedance. **e** Plots showing the extraction of the $C_{dl}$ for various catalysts. **f** Digital photo shows the generated H$_2$ bubbles on $o$-CoSe$_2$|P-modified carbon fiber paper at 250 mV overpotential in sea water. **g** Comparison of onset potential and overpotential for various non-noble-metal HER single catalysts in alkaline electrolytes. Values were plotted from references where they are reported as such (Supplementary Table 1). **h** Chronoamperometric responses ($j \sim t$) recorded on $o$-CoSe$_2$|P at a constant overpotential of 250 mV. Insets: polarization curves (left) and Nyquist plots (right) recorded from $o$-CoSe$_2$|P catalyst before and after 2000 potential cycles. All the measurements were performed in Ar-saturated 1 M KOH (pH ~ 14) and the reported data were iR-compensated

activity of $o$-CoSe$_2$|P catalyst, highlighting the advance of this doping-inducing phase-transition method for accessing high-performance catalysts (Supplementary Fig. 18).

We then recorded electrochemical impedance spectroscopy (EIS) to probe charge transfer process on $o$-CoSe$_2$|P catalyst in 1 M KOH. Nyquist plots in Fig. 3d present that the charge transfer resistance ($R_{ct}$) of $o$-CoSe$_2$|P is 2.2 Ω at 250 mV overpotential, versus 7930 Ω for $c$-CoSe$_2$ and 19.7 Ω for annealed $c$-CoSe$_2$. The smallest $R_{ct}$ of mere 2.2 Ω indicates much promoted kinetics of charge transfer on $o$-CoSe$_2$|P catalyst. Further, double-layer capacitance ($C_{dl}$), which scales roughly with the effective electrochemically active surface area, was measured for studied catalysts (Supplementary Fig. 19). Our results (Fig. 3e) reveal a considerably larger $C_{dl}$ of $o$-CoSe$_2$|P (12.8 mF cm$^{-2}$) compared with $c$-CoSe$_2$ (1.1 mF cm$^{-2}$) and annealed $c$-CoSe$_2$ (5.4 mF cm$^{-2}$), suggesting more accessible active sites created on $o$-CoSe$_2$|P catalyst. We also systematically investigated the synthetic

parameters of $o$-CoSe$_2$|P that affect the HER activity (Supplementary Fig. 20). We further highlight that $o$-CoSe$_2$|P catalyst demonstrates marked HER activities in acidic and neutral electrolytes, such as 0.5 M H$_2$SO$_4$ (pH = 0, Supplementary Fig. 21) and 1 M phosphate-buffered saline (PBS; pH = 7.02, Supplementary Fig. 22), and even seawater (from Gulf Stream in the Gulf of Mexico, pH = 7.94; Fig. 3f and Supplementary Fig. 23). The above energetic and kinetic metrics, including the onset potential and Tafel slope, make $o$-CoSe$_2$|P a superior catalyst to any previously reported noble-metal-free HER single catalysts in alkaline environments (Fig. 3g and Supplementary Table 1). These metrics also outperform those of most HER composite catalysts developed recently (Supplementary Fig. 24 and Supplementary Table 2). Besides activity, another essential factor for real use of a catalyst is the operating stability. We performed aggressive long-term stability tests on $o$-CoSe$_2$|P catalyst by means of chronoampero-metry ($j \sim t$), showing no current decay over 20 h of continuous

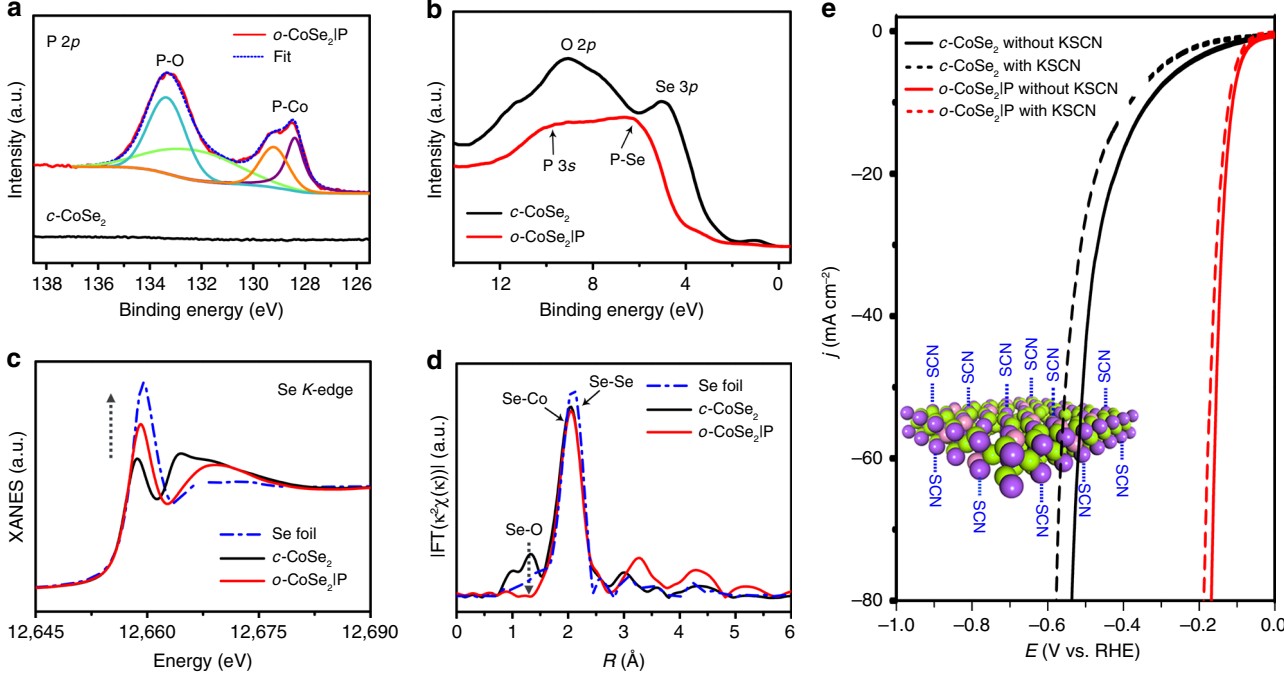

**Fig. 4** Comparison of surface structure of $CoSe_2$ before and after P-doping. **a**, **b** P 2*p* XPS spectra and valence band edge XPS spectra of *c*-$CoSe_2$ and *o*-$CoSe_2|P$ catalysts, respectively. **c**, **d** Se *K*-edge X-ray absorption near-edge spectra and corresponding $k^2$-weighted Fourier transforms spectra for *c*-$CoSe_2$, Se foil, and *o*-$CoSe_2|P$ catalysts. **e** Comparison of $SCN^-$ ions effects on the HER activities of *c*-$CoSe_2$ and *o*-$CoSe_2|P$, respectively. Inset shows the schematic of Co centers blocked by the $SCN^-$ ions

operation at 250 mV overpotential in 1 M KOH (Fig. 3h). Accelerated cyclic voltammetry cycling test further evidences this catalytic robustness with almost no shift of polarization curves after 2000 cycles (inset in Fig. 3h, left), agreeing with EIS measurements, where Nyquist plots exhibits no increase of $R_{ct}$ after cycling (inset in Fig. 3h, right). Various postmortem characterizations reveal no evidence for the structure and phase changes of the cycled sample (Supplementary Fig. 25). We also stress that the HER activity remains undegraded for *o*-$CoSe_2|P$ even after storing it under lab environment for 2 months (storing in an airtight sample tube), presenting its good chemical stability (Supplementary Fig. 26).

**HER enhancement mechanism.** The P-doping-assisted structural phase transition from *c*-$CoSe_2$ to *o*-$CoSe_2|P$ gives a material that efficiently and robustly catalyzes the HER in alkali, exceeding all of the other inexpensive HER single catalysts. We now turn to discuss the structural and phase features that affect the observed properties.

Figure 4a provides the XPS analysis of *o*-$CoSe_2|P$ catalyst in the P 2*p* region, which shows two broad peaks that can be deconvoluted into four bands at ~133.6/132.8 and 129.2/128.3 eV, corresponding to P–O and P–Co bonds[37], respectively. The detection of P–O signal is ascribed to partial surface oxidation of the sample after exposure to air, where the surface oxygen could be removed at the initial stage during the HER process[38]. Further evidence comes from the valence band edge XPS measurements. Figure 4b reveals two peaks at 9.1 eV (O 2*p*) and 5.0 eV (Se 3*p*) for *c*-$CoSe_2$[39], whereas broad peaks at 6.5 eV (P–Se) and 10.1 eV (P 3*s*) were detected for *o*-$CoSe_2|P$[40]. New P–Co and P–Se bonds probed by XPS and valence band edge spectra confirm the participation of P into the structure that aids the phase transition. The disappeared Se 3*p* and O 2*p* signals are the result of P participation that not only bonding with Se for tuned electronic structures but also mitigating the surface oxidation process[41,42].

Our X-ray absorption spectroscopy and EELS analyses demonstrate electron-deficient Co/Se sites and electron-rich P sites in *o*-$CoSe_2|P$ (Fig. 4c, Supplementary Figs. 27–30, and Supplementary Table 3), suggesting that P sites might have high local reactivity for HER than does the *c*-$CoSe_2$. To confirm this, we used thiocyanate ions ($SCN^-$), which are well known to poison the metal-centered catalytic sites[43], to examine its influence on HER activity of *c*-$CoSe_2$ and *o*-$CoSe_2|P$ catalysts. Compared with *c*-$CoSe_2$ whose activity was degraded greatly in the electrolyte with 10 mM $SCN^-$ ions, the negligible HER deactivation for *o*-$CoSe_2|P$ evidences that P sites are effective active sites (Fig. 4e).

Our Se *K*-edge $k^2$-weighted extended X-ray absorption fine structure (Fig. 4d) and Se 3*d* XPS (Supplementary Fig. 31) reveal that Se–O bond is completely suppressed in the *o*-$CoSe_2|P$ catalyst (also see Fig. 4b), which indicates that in accordance with an earlier report[41], the existence of P accounts for the observed marked chemical and catalytic stabilities.

We further carried out density functional theory (DFT) calculations to provide more insights into the remarkable HER property of *o*-$CoSe_2|P$ catalyst (see Methods; Supplementary Figs. 32–34). As compared with *c*-$CoSe_2$, the higher charge density of *o*-$CoSe_2|P$, particularly at P-bonding regions (black arrows), reveals an improved electron environment for catalyzing water electroreduction (Fig. 5a). Evaluation of studied catalysts by calculated hydrogen adsorption free energy ($\Delta G_H$) gave a very small $\Delta G_H$ value of −0.08 eV for *o*-$CoSe_2|P$ at P sites (close to the thermoneutral value of $\Delta G_H = 0$; Fig. 5b), whereas Co sites in *o*-$CoSe_2|P$ show strong water affinity and consequent water dissociation ability (Supplementary Fig. 35), which together imply a synergistic interplay between Co (water adsorption/dissociation) and P (water reduction) that leads to the enhanced energetics for HER. Additionally, the density of states (DOS) results uncover that *o*-$CoSe_2|P$ bears higher states in the characteristic low-DOS region close to the Fermi level (Fig. 5c and Supplementary Fig. 36). Meanwhile, work functions

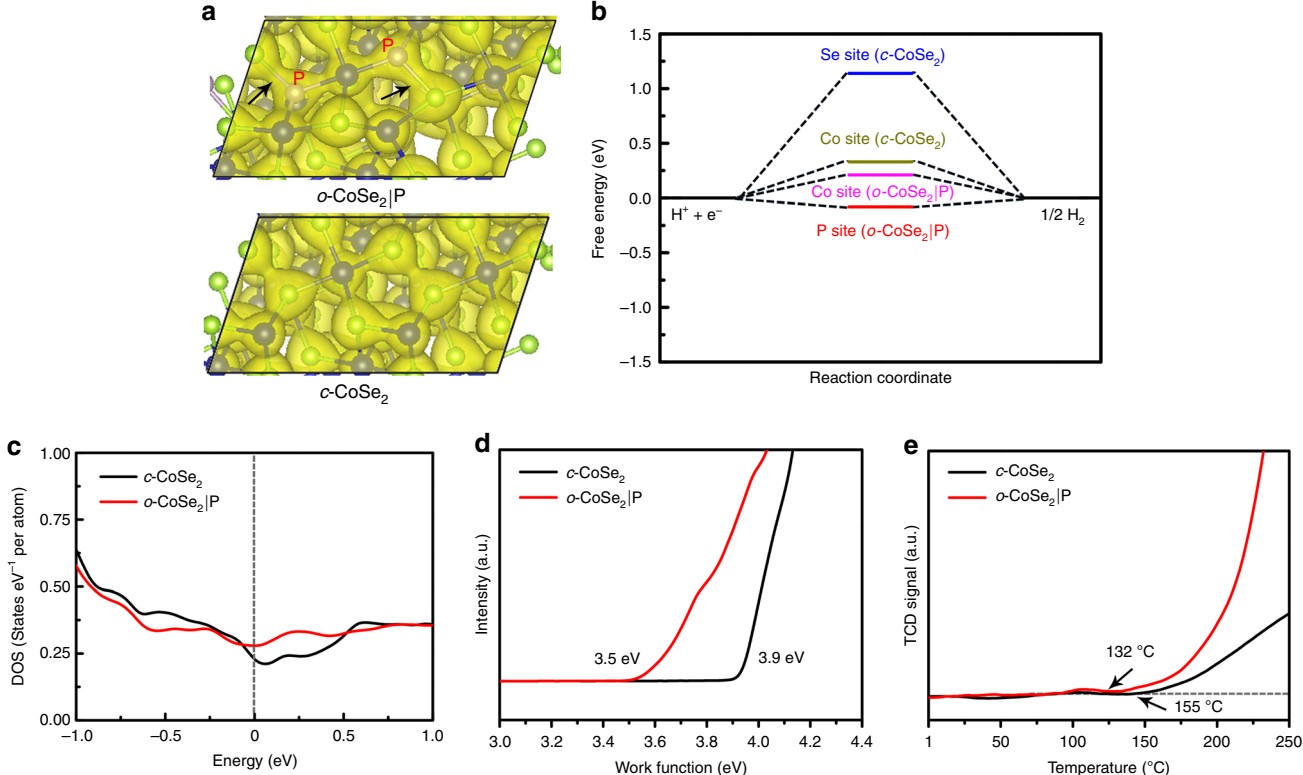

**Fig. 5** DFT calculation and enhancement mechanism. **a** Calculated charge density distribution for $o$-CoSe$_2$|P (up) and $c$-CoSe$_2$ (down) catalysts. **b** Free energy diagrams for hydrogen adsorption at different sits on the (111) surface of $o$-CoSe$_2$|P (8 wt%) and $c$-CoSe$_2$. **c** Calculated total densities of states of $c$-CoSe$_2$ and $o$-CoSe$_2$|P with the Fermi level aligned at 0 eV. **d**, **e** Ultraviolet-photoelectron spectra and H$_2$ temperature-programmed desorption analyses for $c$-CoSe$_2$ and $o$-CoSe$_2$|P catalysts, respectively

measured for $c$-CoSe$_2$ and $o$-CoSe$_2$|P catalysts are 3.9 and 3.5 eV (Fig. 5d), respectively. Such high DOS and low work function for $o$-CoSe$_2$|P associate with promoted electron transfer and enhanced chemical activity. We further evaluated the studied catalysts by temperature-programmed desorption analysis (Fig. 5e), which gave lower H$_2$ onset desorption temperature of 132 °C for $o$-CoSe$_2$|P. We regard this to offer catalytic surfaces for better H$_2$ release, which benefits the Volmer-Heyrovsky mechanism with H$_2$ desorption as the rate-determining step.

## Discussion

In conclusion, we report here that a phosphorus-doping process can induce the structural phase transition from cubic to orthorhombic phases in layered CoSe$_2$. The achieved $o$-CoSe$_2$|P catalyst shows high HER activity and stability in alkaline electrolyte. This remarkable energetics for HER can be explained by the favorable electronic structure and local reactivity that rooted from the phosphorus dopants, which provide optimal binding of reaction intermediates, as confirmed by experimental and computational results. We anticipate that such doping-induced phase-transition method can be extended to other TMD material systems, and thereby promote the development of newly advanced catalysts that make use of Earth-abundant elements to efficiently catalyze desired reactions.

## Methods

**Material synthesis**. All chemicals were used as received without further purification. The $o$-CoSe$_2$|P was prepared through a two-step method. First, layered CoSe$_2$ nanobelts were synthesized as described in our recent work[27]. Then, 50 mg fresh CoSe$_2$ nanobelts and 500 mg of NaH$_2$PO$_2$·H$_2$O were placed at two separated positions in a ceramic boat with the NaH$_2$PO$_2$·H$_2$O at the upstream side. With a heating rate of 5 °C min$^{-1}$, the samples were heated at 400 °C for 30 min in Ar atmosphere. For the time-dependent experiments, the temperature was kept at 400

°C while reactions stopped at desired time; for the temperature-dependent experiments, the reaction time kept at 30 min while the reaction temperature was changed as needed. All the obtained samples were carefully washed and dried before use.

**Characterization**. The achieved samples were examined by various analytical techniques. XRD was performed on a Japan Rigaku DMax-γA X-ray diffractometer with Cu Kα radiation (λ = 1.54178 Å). The morphology of the samples was determined by SEM (Zersss Supra 40) and JEOL 2010F(s) TEM. The STEM and HRTEM images, EELS, SAED, and EDX elemental mappings were taken on JEM-ARM 200F Atomic Resolution Analytical Microscope with an acceleration voltage of 200 kV. Raman spectra were taken on a Raman microscope (Renishaw®) excited with a 514 nm excitation laser. ICP data were obtained by an Optima 7300 DV instrument. The H$_2$-TPD measurements were carried out on AutoChem II 2920. Typically, the samples were pretreated at 250 °C in Ar for 2 h to remove the impurities, which were cooled down to −30 °C and pulse chemisorption of ultrapure H$_2$ at the same time. The temperature was ramped up at 5 °C s$^{-1}$ to 300 °C. The TPD gases were carried out by Ar, detected by thermal conductivity detector. N$_2$ adsorption/desorption isotherms were recorded on an ASAP 2020 accelerated surface area and a porosimetry instrument (Mictromeritics), equipped with an automated surface area, at 77 K using Barrett-Emmett-Teller calculations. Room-temperature electron paramagnetic resonance (EPR) spectra were performed on a JEOL JES-FA200 EPR spectrometer (300 K, 9064 MHz, X band). Ultraviolet-photoelectron spectroscopy was carried out at the BL11U beamline of National Synchrotron Radiation Laboratory in Hefei, China. The X-ray absorption spectra of Co and Se K-edges were obtained at the beamline 14W1 of Shanghai synchrotron Radiation Laboratory (China), while the P K-edges were performed at the beamline 4B7A station of Beijing Synchrotron Radiation Facility (China).

**Electrochemical measurements**. All the electrochemical measurements were performed in a conventional three-electrode cell at ambient temperature connected to a Multipotentiostat (IM6ex, ZahnerElectrik, Germany). Saturated calomel electrode and graphite rod were used as the reference and counter electrodes, respectively. The potentials reported in this work were normalized versus the RHE through a standard RHE calibration described elsewhere[8]. A rotating disk electrode (RDE) with glassy carbon (PINE, 5.00 mm diameter, disk area: 0.196 cm$^2$) was used as the working electrode in performing the HER activity, an RRDE with both a glassy carbon disk (5.61 mm diameter, disk area: 0.2475 cm$^2$) and a Pt ring (6.25

mm inner-diameter and 7.92 mm outer diameter, ring area: 0.1866 $cm^2$) was used for confirming the $H_2$ evolution. For the stability tests, a graphite rod was used as the counter electrode to avoid the possible contribution of dissolved Pt species during HER.

To make the working electrodes, 5 mg catalyst powder was dispersed in 1 ml of 1:3 v/v isopropanol/DIW mixture with 40 μl Nafion solution (5 wt%), which was ultrasonicated for ~30 min to yield a homogeneous ink. Then, a certain volume of dispersion was pipetted onto the glassy carbon substrate, resulting in catalyst loading of ~1.02 mg $cm^{-2}$. HER measurements were conducted in 0.5 M $H_2SO_4$, 1.0 M KOH, 1.0 M PBS (pH = 7.02), and nature seawater (pH = 7.84, Gulf of Mexico, Gulf Stream of Dauphin Island, Alabama), respectively. All the fresh electrolytes were bubbled with pure argon for 30 min before measurements. The polarization curves were obtained by sweeping the potential from −0.55 to 0.1 V versus RHE with a sweep rate of 10 mV $s^{-1}$ and 1600 r.p.m (to remove the $H_2$ bubbles formed in situ) at ambient temperature. The EIS measurement was performed in the same configuration at 250 mV overpotential over a frequency range from 100 KHz to 100 mHz at the amplitude of the sinusoidal voltage of 5 mV and room temperature. The Pt-ring electrode of RRDE was kept at 0.5 V versus RHE during HER to detect the produced $H_2$ at the disc electrode. The polarization curves were replotted as overpotential ($\eta$) versus log current (log $j$) to get Tafel plots for assessing the HER kinetics of investigated catalysts. The Tafel slope ($b$) can be obtained by fitting the linear portion of the Tafel plots with the following equation,

$$\eta = b \log(j) + a \qquad (1)$$

The influence of $SCN^-$ on the HER activity of investigated catalysts was evaluated by adding 10 mM $SCN^-$ in the electrolyte. The o-$CoSe_2$|P-modified carbon fiber paper (catalyst loading: 1 mg $cm^{-2}$) was used as working electrode to perform chronoamperometry experiments at 250 mV overpotential.

The accelerated stability measurements were performed by potential cycling between −0.4 and −0.1 V versus RHE with a sweep rate of 100 mV $s^{-1}$. After cycling, the resultant electrode was used for polarization curves with a sweep rate of 10 mV $s^{-1}$. To estimate the double-layer capacitance, cyclic voltammograms were performed at different sweep rates in the potential region of −0.1–0 versus RHE at ambient temperature. All the polarization curves were corrected with iR compensation that arised from the solution resistance. We employed ICP method to analyze the etching rate of o-$CoSe_2$|P during HER process in 1 M KOH at a constant 250 mV overpotential. The catalyst was loaded on the $1 \times 1.5$ $cm^2$ carbon paper substrate (~1.0 mg $cm^{-2}$). Each ICP point was collected for three times. The gas production of $H_2$ evolution was monitored by gas chromatography (GC2014, Shimadzu, Japan) equipped with a TCD detector with argon as a carrier gas.

**DFT calculations**. We carried out DFT calculations using the Vienna ab initio simulation package. The exchange-correlation energy was described using the Perdew-Burke-Ernzerhof. A 280 eV plane-wave kinetic energy cutoff was chosen, and a $3 \times 5 \times 1$ Monhorst-Pack $k$-point sampling was adopted for the structure relaxation. A residual force threshold of 0.02 eV $Å^{-1}$ was set for geometry optimizations. Details of the calculation are provided in the Supplementary Figs. 32–36, Supplementary Table 4, and Supplementary Note 2.

**Data availability**. The data that support the findings of this study are available on request from the corresponding authors (M.-R.G. or S.-H.Y.).

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

## Acknowledgements

We acknowledge the funding support from the National Natural Science Foundation of China (Grants 21431006, 21761132008, 51702312, and 21703228), the Foundation for Innovative Research Groups of the National Natural Science Foundation of China (Grant 21521001), Key Research Program of Frontier Sciences, CAS (Grant QYZDJ-SSW-SLH036), the National Basic Research Program of China (Grant 2014CB931800), the Users with Excellence and Scientific Research Grant of Hefei Science Center of CAS (2015HSC-UE007), the Fundamental Research Funds for the Central Universities (WK2340000076), and the Recruitment Program of Global Youth Experts. Y.-R.Z. acknowledges the China Postdoctoral Science Foundation (2016M592063). This work was partially carried out at the USTC Center for Micro and Nanoscale Research and Fabrication.

## Author contributions

Y.-R.Z. planned and performed the experiments, collected and analyzed the data, and wrote the paper. M.-R.G. and S.-H.Y. supervised the project, conceived the experiments, analyzed the results, and wrote the paper. P.W. and Z.L. performed the DFT calculations. R.Y. and W.-X.H. performed the H$_2$-TPD measurements. S.-J.L. performed the XANES and EXAFS experiments. H.-X.J., S.-W.H., and J.F.Z. performed XPS and UPS measurements. X.-L.Z., R.W., F.-Y.G., and Q.G. assisted with the experiments and characterizations. All authors discussed the results and commented on the manuscript.

## Additional information

**Competing interests:** The authors declare no competing interests.

