## [Peer Review File · Nature Communications]

Reviewer #1 (Remarks to the Author):

This work presents an interesting phosphorus doping induced structural phase transition of cubic CoSe₂ (c-CoSe₂) to orthorhombic CoSe₂ (o-CoSe₂). The authors systematically investigated influence of the experimental parameters (e.g. time, temperature, dopant amount) on the structure and phase of the CoSe₂ and carefully explored the phase transition mechanism, which provides insights in to the phase engineering of CoSe₂ (and possibly other transition metal dichalcogenides). The achieved phosphorus o-CoSe₂ (o-CoSe₂|P) exhibits high electrocatalytic activity towards the hydrogen evolution reaction (HER) with a low overpotential of 104 mV at 10 mA cm⁻² in 1 M KOH. The catalyst is found also to efficiently catalyze HER in neutral media. The authors further studied the high activity of o-CoSe₂|P by through broad characterizations and theoretical calculations. In general, this is a decent work and I recommend its publication in Nat. Commun. after the authors address/clarify the following points.

1. The high HER activity of o-CoSe₂|P is mainly attributed to the P active sites as revealed by the DFT calculation. It would be better to also confirm it experimentally by comparing the HER performance of o-CoSe₂|P and pure phase o-CoSe₂. I am also curious that if o-CoSe₂ can be doped by P through such method and what performance would be?
2. The DFT calculation indicates the Co sites have a strong water affinity whereas P sites possess a small hydrogen adsorption free energy. While one of the important step of HER in alkaline solution is water dissociation. The authors should further calculate the adsorption free energy of the activated water (HO⁻ - H) on the catalyst.
3. Phosphide materials generally exhibit high HER activity and stability in acid but are much less stable in alkaline solutions (e.g. JACS 2013, 135, 9267; PCCP 2014, 16, 5917). The P in o-CoSe₂|P are partially in the form of phosphide (Co-P) therefore it would be better to monitor the etching rate of the material (Co and P) during the HER by ICP or other suitable technique. If no Co/P species can be detected in the electrolyte, then the authors can report an upper limit to the etching rate. The post-HER characterization such as XPS is also recommend to further support the stability claim.
4. Have the authors tested the HER activity of o-CoSe₂|P in acid? Chalcogenides are typical acid-stable HER catalysts with high performance.
5. The system resistance seems quite high (i.e. more than 5 ohm in 1 M KOH and more than 15 ohm in neutral electrolytes). Such high resistance would result in a significant potential shift when the IR-correction is applied. Please present the original polarization curves without IR correction and please indicate the method used for IR correction.
6. Table S1 and S2 should also include the mass loading and the electrochemically active surface area for each sample where-ever possible. The authors used a high mass loading of ~1.02 mg cm⁻², which is why the performance seems good. The typical area density of electrocatalysts on GCE is ~200 μg cm⁻², thus it is not accurate for the comparison with those catalysts on GCE. These tables should also include Ni-Mo catalysts for comparison as Ni-Mo alloys are the benchmarking catalysts in alkaline electrolytes.
7. With respect to point 6 above, the authors should also report the surface area normalized exchange current density. The exchange current density only meaning with respect to the intrinsic activity when the microscopic surface area is accounted for.
8. Please calculate the Faradaic efficiency of the hydrogen production.

Reviewer #2 (Remarks to the Author):

The authors demonstrated a phosphorus doping-induced phase transition of CoSe₂ from cubic phase to orthorhombic phase and characterized the transition with various techniques including XRD, Raman, and XPS. o-CoSe₂|P with ~8 wt% of P dopant was shown to have good HER activity and stability in 1 M KOH, with 104 mV overpotential at 10 mA/cm² and no obvious degradation in activity after 2-month storage under lab environment or 2000 electrochemical cycles. Experimental results and DFT calculation revealed that P sites possess appropriate hydrogen adsorption-desorption energy and could serve as extra active sites, whereas Co sites show strong water affinity, together rendering a synergistic effect between the dopant and o-CoSe₂ matrix in catalyzing HER. The manuscript is recommended for acceptance after the following comments/questions are addressed by the authors.

Comments on experiments, calculations, and discussions:

1. In Figure 2d, c-CoSe₂ has a representative Raman peak at 189 cm⁻¹ while o-CoSe₂|P shows a peak at 174 cm⁻¹. The authors described such difference as a shift of Raman peak from 189 cm⁻¹ to 174 cm⁻¹ resulting from increased Se-Se (P) bond length in the converted o-CoSe₂|P. However, since cubic and orthorhombic CoSe₂ are in different crystal structures, they would have different symmetry and vibrational modes. The two different peaks may arise from two different Raman-active vibrational modes and thus cannot be compared side by side.
2. Figure 4a reveals the P 2p spectrum of o-CoSe₂|P and P-Se bonding was de-convoluted from the broad peak positioned at around 133 eV – 134 eV to assist the authors' claim on P participation in the o-CoSe₂ structure. However, in Ref. 38 cited, only PO₄³⁻ is mentioned but not the P-Se bonding. It is inappropriate to perform peak fitting/deconvolution without the basis on solid scientific reference.
3. Figure 2b and 4a display the XPS spectra of P in o-CoSe₂|P. Both the P 2s and P 2p spectra show dominant peaks arising from oxidized P. In the DFT calculation, only reduced P (Co-P bonds) was considered. How will such oxidized species affect the HER performance of o-CoSe₂|P? A recent publication reveals that oxidized species on CoP nanoparticle surface will be reduced under HER conditions in 1 M KOH (*J. Phys. Chem. C*, **2018**, *122*, 2848). If the oxidized species on o-CoSe₂|P surface had similar responses under HER conditions, the simulation with perfectly clean and non-oxidized surface of o-CoSe₂|P might be fine. Otherwise it would be problematic.
4. The enhancement of HER activity after P-doping induced phase conversion was very substantial (Figure 3a). The authors should compare o-CoSe₂|P with bare o-CoSe₂ without doping to unravel the different dependency of HER performance on crystal structure and P-doping.
5. These two highly related publications should be cited: *ACS Catalysis*, **2017**, *7*, 4026; *J. Phys. Chem. C*, **2018**, *122*, 2848

Comments on non-scientific aspects:

1. In Figure 2a, Ar-400 °C should correspond to cubic-CoSe₂ annealed at 400 °C without P precursor. However, putting together with other notations such as 300 °C and 400 °C without defining the notation in the main text would be ambiguous. Please check that all notations throughout the manuscript are defined and clarified.
2. Figure 2b was mentioned in the main text as P 2p XPS by mistake. It should be P 2s spectra as described in the caption.
'These results match well with P 2p XPS and EDX data (Figs. 2b and c), taking into account that gradual increase in P content is responsible for yielding o-CoSe₂|P.'
3. In the calculation of TOF in supporting information,

$$\frac{0.01 A}{1 \text{ cm}^2} \times \frac{1 C}{1 s} \times \frac{1 \text{ mol}}{96485 C} \times \frac{6.02 \times 10^{23} e}{1 \text{ mol}} \times \frac{1}{2 e} \times \frac{1 \text{ cm}^2}{4.81 \times 10^{16} \text{ atom}} = 0.65 \frac{H_2/s}{\text{surface site}}$$

the second term should be $\frac{1 C}{1 s * 1 A}$ to balance the dimensions on both sides.

- Figure S23 in supporting information manifests that after 2 months of storage in laboratory, o-CoSe₂|P still possessed high HER activity. It would be more specific to indicate what the 'lab environment' is, whether it was stored in a vacuumed desiccator or in a vial sitting in the air, etc.

Reviewer #3 (Remarks to the Author):

The manuscript focuses on an interesting approach to develop a novel P-doped o-CoSe₂ catalyst for the HER in alkaline media. The manuscript is well-written. The electrochemical performance of the catalysts is reasonably good, although the activity (104 mV@10 mA/cm²) is unfortunately not comparable to those in the highest activity catalyst group in literature. I regret to say that the manuscript contains some technically insufficient/inappropriate issues. I consider that the present manuscript will be reconsidered for a publication to Nature Communications after revision. The following is my comments.

(1) There are no descriptions on experimental setups/procedures for the temperature-dependent XRD, XPS, EDX, and Raman measurements (from line 111 in page 7 & Figure 2). Are they in situ measurements during annealing?

(2) Line 125 in page 8: "Raman spectrum also uncovers..." There is no Raman spectrum shown in manuscript?

(3) Specify how the authors define a threshold for the onset potential (line 152 in page 9).

(4) Line 232 in page 13. "whereas only one broad peak at 6.5 eV(P-Se)..." What the broad peak around 10 eV represents, O?

(5) XAS data (Figures 4s&d, Figure S4): Plot data from Co and Se references together. Which features in Figures 4s&d and Figure s24 represent "electron-deficient Co/Se sites and electron-rich P sites in o-CoSe₂-P" (line 237 in page 13)? Fitting of EXAFS should be done concurrently using Se K edge and Co K edge data, and evaluate coordination numbers (CNs) of Co-Co, Co-Se, Se-Se, and Se-Co, as well as bonding distances, and make sure if CNs of Co-Co and Se-Se show low values (close to zero). For example, see a ref (A. Frenkel, Solving the 3D structure of metal nanoparticles, Z. Kristallogr. 222, 605-611 (2007)).

(6) English needs to be refurbished.

We thank all the reviewers for their valuable comments and questions that help us significantly improve the revised manuscript.

Reviewer #1 (Remarks to the Author):

This work presents an interesting phosphorus doping induced structural phase transition of cubic CoSe₂ (c-CoSe₂) to orthorhombic CoSe₂ (o-CoSe₂). The authors systematically investigated influence of the experimental parameters (e.g. time, temperature, dopant amount) on the structure and phase of the CoSe₂ and carefully explored the phase transition mechanism, which provides insights in to the phase engineering of CoSe₂ (and possibly other transition metal dichalcogenides). The achieved phosphorus o-CoSe₂ (oCoSe₂|P) exhibits high electrocatalytic activity towards the hydrogen evolution reaction (HER) with a low overpotential of 104 mV at 10 mA cm⁻² in 1 M KOH. The catalyst is found also to efficiently catalyze HER in neutral media. The authors further studied the high activity of o-CoSe₂|P by through broad characterizations and theoretical calculations. In general, this is a decent work and I recommend its publication in Nat. Commun. after the authors address/clarify the following points.

Response: We greatly appreciate the reviewer for the positive feedbacks on our work.

1. The high HER activity of o-CoSe₂|P is mainly attributed to the P active sites as revealed by the DFT calculation. It would be better to also confirm it experimentally by comparing the HER performance of o-CoSe₂|P and pure phase o-CoSe₂. I am also curious that if o-CoSe₂ can be doped by P through such method and what performance would be?

*Response: We thank the reviewer for the thoughtful suggestions. Following your suggestion, we made P-doped o-CoSe₂ via a similar annealing strategy but using o-CoSe₂ to replace c-CoSe₂ (see **Supplementary Figures 18a and b**). To ensure a fair comparison, we carefully tuned the P doping content to around 7.75 wt% (see **Supplementary Figure 18c**), which is very close to that of 8 wt% for o-CoSe₂|P. We thus studied the HER performances of pure o-CoSe₂, P-doped o-CoSe₂ and o-CoSe₂|P in Ar-saturated 1 M KOH. As demonstrated clearly in **Supplementary Figure 18d**, although P-doped o-CoSe₂ shows good activity promotions as compared with pure o-CoSe₂, it is still inferior to the performance of the new o-CoSe₂|P catalyst. We attribute this observation to the doping-induced bonds rotation and structure reconstructions, which create additional surface sites that are HER active, further highlighting the effectiveness of this doping-inducing phase-transition method for accessing newly advanced electrocatalysts. These results have been added to the revised Supplementary Information as **Supplementary Figure 18** with corresponding discussion provided in the revised Manuscript.*

2. The DFT calculation indicates the Co sites have a strong water affinity whereas P sites possess a small hydrogen adsorption free energy. While one of the important step of HER in alkaline solution is water dissociation. The authors should further calculate the adsorption free energy of the activated water (HO- - H) on the catalyst.

Response: We thank the reviewer for the thoughtful suggestion. On the basis of this suggestion, we performed further DFT calculations regarding the water dissociation at the Co, Se and P sites of *o*-CoSe₂/P. As revealed in **Supplementary Figure 35b**, H₂O moleculars are found to be hardly adsorbed on the Se and P sites, making that the dissociated energy of H₂O on the two sites is similar in vacuum. But when we switched to the Co site of *o*-CoSe₂/P, we observed that the energy barrier of H₂O dissociation largely reduced from 4.40 eV (in vacuum) to mere 0.50 eV, suggesting that Co sites not only bear a strong water affinity, but also easily activate H₂O to generate intermediate H and OH species. Hence, our new calculations further evidence the synergistic interplay between Co (water adsorption and dissociation) and P (water reduction) that gives rise to the remarkable HER energetics. We added this new computational result in the revised SI as **Supplementary Figure 35b**.

3. Phosphide materials generally exhibit high HER activity and stability in acid but are much less stable in alkaline solutions (e.g. JACS 2013, 135, 9267; PCCP 2014, 16, 5917). The P in *o*-CoSe₂/P are partially in the form of phosphide (Co-P) therefore it would be better to monitor the etching rate of the material (Co and P) during the HER by ICP or other suitable technique. If no Co/P species can be detected in the electrolyte, then the authors can report an upper limit to the etching rate. The post-HER characterization such as XPS is also recommend to further support the stability claim.

Response: We would like to thank the reviewer for the critical comments and suggestions. This was also our No. 1 concern because if the Co and P were etched away from the structure, the active sites on *o*-CoSe₂/P will be removed. We therefore made significant efforts to make sure whether it is structure stable in alkali.

We first address the concern of the structure stability by employing ICP measurements to monitor the etching amounts of Co and P during the HER process in 1M KOH with a 250 mV overpotential. The *o*-CoSe₂/P catalyst was uniformly loaded onto a 1×0.5 cm² carbon paper substrate (~1.0 mg cm⁻²). Our ICP results (three measurements for each data point) show little concentration changes of Co and P in the electrolyte (see **Supplementary Figure 25f**), which indicates the marked structure stability of *o*-CoSe₂/P even in alkali, agreeing with our stability test that the HER activity of *o*-CoSe₂/P was maintained after long-term operation. We regard this impressive stability to the doping-induced structure reconstruction, which keeps the structure intact in alkali, quite different from previous observations.

We also used XPS technique to study the surface chemistry of cycled *o*-CoSe₂/P catalyst. It reveals that no obvious chemical state changes for both Co and P over 20 h of continuous operation (see **Supplementary Figures 25g and h**; **Note:** the broad peak at ~134.0 eV comes from the P-O bond, ascribing to the partial surface oxidation when taking the cycled sample out of the electrolyte for XPS measurement).

Therefore, on the basis of our new ICP and XPS results, we again confirm the

outstanding structure stability of the newly developed o-CoSe₂|P catalyst.

4. Have the authors tested the HER activity of o-CoSe₂|P in acid? Chalcogenides are typical acid-stable HER catalysts with high performance.

Response: *We thank the reviewer's comment and suggestion. In this work we mainly want to report the P-doping induced structural phase transition strategy that creates a new doped-TMD material with high HER properties in alkaline electrolytes, in which the HER kinetics are two orders of magnitude slower than that in acid on Pt, thus new low-cost catalyst systems need to be developed. Following your suggestion, we have also investigated the HER properties of o-CoSe₂|P in Ar-saturated 0.5 M H₂SO₄ at ambient temperature, which demonstrates a good HER activity with a low overpotential of ~100 mV at 10 mA cm⁻², as well as a impressive HER stability (see **Supplementary Figure 21**). We have added these new data as **Supplementary Figure 21** in the revised SI for readers' information.*

5. The system resistance seems quite high (i.e. more than 5 ohm in 1 M KOH and more than 15 ohm in neutral electrolytes). Such high resistance would result in a significant potential shift when the *iR*-correction is applied. Please present the original polarization curves without *iR* correction and please indicate the method used for *iR*-correction.

Response: *We thank the reviewer's comment and suggestion. We used the EIS technique to gain the solution resistance (R_s). In the EIS Nyquist plot, the point where the first intercept of the main arc with the real axis stands for the R_s . Our measured R_s values of 5 ohm in 1M KOH and 15 ohm in neutral electrolyte matches well with prior reports of ~6 ohm in 1 M KOH (Nat. Commun., 2015, 6, 6512) and ~15 ohm in 1 M PBS (Chem. Commun., 2015, 51, 4252). To do the *iR*-correction, we used the equation $E_{corrected} = E_{raw} - iR_s$ and then plotted it with current density. We provided the original polarization curves without *iR*-correction and the *iR*-corrected curves for comparison purpose, as shown in Figure Q1.*

Figure Q1. iR -corrected and the original HER polarization curves of different catalysts in alkaline (a, b) and neutral media (c, d).

6. Table S1 and S2 should also include the mass loading and the electrochemically active surface area for each sample where-ever possible. The authors used a high mass loading of $\sim 1.02 \text{ mg cm}^{-2}$, which is why the performance seems good. The typical area density of electrocatalysts on GCE is $\sim 200 \text{ } \mu\text{g cm}^{-2}$, thus it is not accurate for the comparison with those catalysts on GCE. These tables should also include Ni-Mo catalysts for comparison as Ni-Mo alloys are the benchmarking catalysts in alkaline electrolytes.

Response: Thanks for the thoughtful suggestions. We have added the mass loadings of different catalysts in the Tables for comparison (see revised **Supplementary Tables 1-2**). But for the electrochemical active surface area (ECSA), none of previous reports provides such parameter. The reason is that ECSA values are difficult to obtain for these materials: they cannot be calculated using the classic hydrogen under-potential deposition (UPD) like commonly done for Pt because no obvious hydrogen adsorption occurs prior to H_2 evolution; they cannot be calculated using capacitance ratio method because the lack of their specific capacitances. Thus in this case, we are sorry that we are unable to offer ECSA values of these catalysts for comparison.

We surveyed the mass loading adopted for different catalysts in literatures, and found that the loading is really diverse. For examples, high loadings of 43.4 mg cm^{-2} , 25.8 mg cm^{-2} , 4.0 mg cm^{-2} , 2.62 mg cm^{-2} , 2.1 mg cm^{-2} and others were reported for different catalysts, including on bulk supports and GCE (see revised **Supplementary Tables 1-2**). Therefore a comparison on the basis of the same loading is unlikely to do.

For o-CoSe₂/P catalyst, we observed that the loading of ~1.02 mg cm⁻² gives the optimal HER activity, even exceeding most of the reported catalysts with higher loadings, which clearly demonstrate the advance of the new o-CoSe₂/P catalyst.

*Following up your suggestion, we now include the Ni-Mo-based catalysts reported in recent years for comparison, as shown in revised **Supplementary Tables 1-2**.*

7. With respect to point 6 above, the authors should also report the surface area normalized exchange current density. The exchange current density only meaning with respect to the intrinsic activity when the microscopic surface area is accounted for.

Response: *We thank the reviewer for the helpful suggestion. As described above, the best way to evaluate the intrinsic electrochemical activity of a catalyst is to calculate its specific activity based on its electrochemically active surface area (ECSA), and thus other effects such as morphologies, sizes and pores can be eliminated. However, the ECSA value are difficult to obtain for o-CoSe₂/P catalyst: it cannot be calculated using the classic hydrogen under-potential deposition (UPD) like commonly done for Pt because no obvious hydrogen adsorption occurs prior to H₂ evolution; it cannot be calculated using capacitance ratio method because the lack of their specific capacitances. However, the intrinsic electrocatalytic activities of o-CoSe₂/P catalyst can be reasonably compared by using their BET surface areas. According to Trasatti et al.'s work (Pure & Appl. Chem., **1991**, 63, 711-734), BET surface area is a very effective method to replace ECSA when ECSA is difficult or impossible to obtain. Based on your comments, we measured the BET surface areas of studied catalysts and offered the BET normalized exchange current density, which again evidence the high intrinsic activity of the new o-CoSe₂/P catalyst. We added the new data as **Supplementary Figure 15** in the revised SI.*

8. Please calculate the Faradaic efficiency of the hydrogen production.

Response: *Thanks for the good suggestion. We performed a long-term electrolysis on o-CoSe₂/P electrode at 10 mA cm⁻², the H₂ generation was carefully analyzed via gas chromatography. The detected amount of H₂ gas is consistent with the theoretical value, corresponding to a Faradaic efficiency of close to 100 %, as shown in our new data (**Supplementary Figure 17**) in the revised SI.*

Reviewer #2 (Remarks to the Author):

The authors demonstrated a phosphorus doping-induced phase transition of CoSe₂ from cubic phase to orthorhombic phase and characterized the transition with various techniques including XRD, Raman, and XPS. o-CoSe₂/P with ~8 wt% of P dopant was shown to have good HER activity and stability in 1 M KOH, with 104 mV overpotential at 10 mA/cm² and no obvious degradation in activity after 2-month storage under lab environment or 2000 electrochemical cycles. Experimental results and DFT calculation revealed that P sites possess appropriate hydrogen adsorption-desorption energy and could serve as extra active sites, whereas Co sites

show strong water affinity, together rendering a synergistic effect between the dopant and o-CoSe₂ matrix in catalyzing HER. The manuscript is recommended for acceptance after the following comments/questions are addressed by the authors. Comments on experiments, calculations, and discussions:

Response: *We greatly appreciate the reviewer's highly positive feedbacks and strong support on our work.*

1. In Figure 2d, c-CoSe₂ has a representative Raman peak at 189 cm⁻¹ while o-CoSe₂|P shows a peak at 174 cm⁻¹. The authors described such difference as a shift of Raman peak from 189 cm⁻¹ to 174 cm⁻¹ resulting from increased Se-Se (P) bond length in the converted o-CoSe₂|P. However, since cubic and orthorhombic CoSe₂ are in different crystal structures, they would have different symmetry and vibrational modes. The two different peaks may arise from two different Raman-active vibrational modes and thus cannot be compared side by side.

Response: *We thank the reviewer for the insightful comments. These comments motivated us to further study the Raman properties of as-obtained materials. We note that both pyrite and orthorhombic structures possess two different center of inversion, including the metal sites and the midpoint of the X₂ pairs. Metal atoms are located at the centers of symmetry, which would keep immobile in all of the Raman active modes. Therefore, the Raman responses of these structures are generated from the opposite movement of the two X atoms with equal amplitudes (i.e., A_g mode: X-X stretching or E_g mode: X₂ libration). It is confirmed that the peak at ca. 190 cm⁻¹ corresponds to the A_g (Se-Se stretching) mode of c-CoSe₂ (J. Chem. Phys., **1976**, 64, 3604). Although the peak at ca. 174 cm⁻¹ could be assigned to o-CoSe₂ (Physica B, **2002**, 324, 409), we are unable to confirm the exact Raman mode yet. We have modified the Figure 2d and related descriptions in the revised MS.*

2. Figure 4a reveals the P 2p spectrum of o-CoSe₂|P and P-Se bonding was de-convoluted from the broad peak positioned at around 133 eV-134 eV to assist the authors' claim on P participation in the o-CoSe₂ structure. However, in Ref. 38 cited, only PO³⁻ is mentioned but not the P-Se bonding. It is inappropriate to perform peak fitting/deconvolution without the basis on solid scientific reference.

Response: *We thank the reviewer for the careful comments. To come to a clearer conclusion, we re-measured the P 2p XPS spectrum of o-CoSe₂|P and re-visited the Ref. 38. The new fitting results shown in revised Figure 4a reveal that the broad peaks at ~133.6/132.8 corresponds to P-O bands because partial surface oxidation of the sample after exposure to air. The formation of P-Se bond can be uncovered by our valence band edge XPS spectra in Figure 4b. In the revision, we have corrected this error and modified related descriptions and Figures.*

3. Figure 2b and 4a display the XPS spectra of P in o-CoSe₂|P. Both the P 2s and P 2p spectra show dominant peaks arising from oxidized P. In the DFT calculation, only reduced P (Co-P bonds) was considered. How will such oxidized species affect the HER performance of o-CoSe₂|P? A recent publication reveals that oxidized species on

CoP nanoparticle surface will be reduced under HER conditions in 1 M KOH (*J. Phys. Chem. C*, **2018**, *122*, 2848). If the oxidized species on o-CoSe₂|P surface had similar responses under HER conditions, the simulation with perfectly clean and nonoxidized surface of o-CoSe₂|P might be fine. Otherwise it would be problematic.

Response: We completely understand and appreciate the reviewer's concern about the surface-P oxidation. This was our first concern when we performed the DFT calculation. We came to the conclusion that the surface of o-CoSe₂|P during HER process is not oxidized based on the following facts. On the one hand, a large body of literatures evidences that the surface oxidized species of a catalyst will be reduced at reduction conditions, such as the oxide-derived nanocrystalline Cu (*Nature*, **2014**, *508*, 504), MoP (*Angew. Chem. Int. Ed.*, **2014**, *53*, 14433) and MoSP (*Adv. Mater.* **2016**, *28*, 1427). Moreover, the in-situ study performed by Wang and co-workers clearly observed the reduction of surface oxidized species on CoP (*J. Phys. Chem. C*, **2018**, *122*, 2848). We also attempted to conduct an in-situ XPS characterization, but it is currently hard for us to do this. On the basis of the observations from the above literatures, we believe that the oxidized species will not exist during HER process. On the other hand, we purchased some black phosphorous (BP) powder and put it in the air to oxidize its surface (**Figure Q2a and b**). We found that the surface oxidized BP is almost HER inactive in 1M KOH. This result means that even if there is few oxidized surface-P is not reduced, it has no contribution to the high HER activity of o-CoSe₂|P catalyst. Taken together, we are confident that the surface of o-CoSe₂|P during the HER process is not oxidized and our DFT model is reasonable.

Figure Q2. (a, b) TEM image and P 2p XPS spectrum of air-oxidized BP, respectively. (c) Comparison of the HER performances of air-oxidized BP and o-CoSe₂|P catalyst in 1 M KOH.

4. The enhancement of HER activity after P-doping induced phase conversion was very substantial (Figure 3a). The authors should compare o-CoSe₂|P with bare o-CoSe₂ without doping to unravel the different dependency of HER performance on crystal structure and P-doping.

Response: We thank the reviewer for the thoughtful suggestions. Following your suggestion, we have compared the HER activities of bare o-CoSe₂ with the new o-CoSe₂|P. As shown in **Supplementary Figure 18d**, the o-CoSe₂|P overwhelmingly exceeds the performance of bare o-CoSe₂. We further made P-doped o-CoSe₂ via a similar annealing strategy but using o-CoSe₂ to replace c-CoSe₂ (see **Supplementary Figure 18a and b**). To ensure a fair comparison, we carefully tuned the P doping

content to around 7.75 wt% (see **Supplementary Figure 18c**), which is very close to that of 8 wt% for *o*-CoSe₂/P. We then studied the HER performances of bare *o*-CoSe₂, P-doped *o*-CoSe₂ and *o*-CoSe₂/P in Ar-saturated 1 M KOH. As demonstrated clearly in **Supplementary Figure 18d**, although P-doped *o*-CoSe₂ shows good activity promotions as compared with bare *o*-CoSe₂, it is still inferior to the performance of the new *o*-CoSe₂/P catalyst. We attribute this observation to the doping-induced bonds rotation and structure reconstructions, which create additional surface sites that are HER active, further highlighting the effectiveness of our doping-inducing phase-transition method for accessing newly advanced electrocatalysts. These results have been added to the revised Supplementary Information as **Supplementary Figure 18** with corresponding discussion provided in the revised Manuscript.

5. These two highly related publications should be cited: *ACS Catalysis*, **2017**, 7, 4026; *J. Phys. Chem. C*, **2018**, 122, 2848

Response: We thank the reviewer for this kind suggestion. These suggested references have been added to the revised Manuscript.

Reviewer #3 (Remarks to the Author):

The manuscript focuses on an interesting approach to develop a novel P-doped *o*-CoSe₂ catalyst for the HER in alkaline media. The manuscript is well-written. The electrochemical performance of the catalysts is reasonably good, although the activity (104 mV@10 mA/cm²) is unfortunately not comparable to those in the highest activity catalyst group in literature. I regret to say that the manuscript contains some technically insufficient/inappropriate issues. I consider that the present manuscript will be reconsidered for a publication to Nature Communications after revision. The following is my comments.

Response: We greatly appreciate the reviewer's positive feedbacks, especially the reviewer's favorable comments on the novelty of our work.

(1) There are no descriptions on experimental setups/procedures for the temperature-dependent XRD, XPS, EDX, and Raman measurements (from line 111 in page 7 & Figure 2). Are they in situ measurements during annealing?

Response: We thank the reviewer for the thoughtful comments. We note that these characterizations are not in-situ/operando measurements, and all these data were collected after annealing at designed temperatures. We are sorry about this confusion and we have modified the related experimental details in the revised MS.

(2) Line 125 in page 8: "Raman spectrum also uncovers..." There is no Raman spectrum shown in manuscript

Response: We thank the reviewer for careful reading our paper. We presented the Raman spectra as Figure 2d in the MS, which clearly show the phase transition from *c*-CoSe₂ to *o*-CoSe₂ through the P-induced annealing process. We have revised related words to make the descriptions clearer to follow.

(3) Specify how the authors define a threshold for the onset potential (line 152 in page 9).

Response: We appreciate the reviewer for the good suggestion. The onset potential is the potential at which the HER current occurs, but in the literatures there is no clear definition regarding the threshold for the onset potential. In this work, we determined the onset potential as the overpotential at which the current density is as small as 1 mA cm^{-2} . We note that we can not use smaller current density ($<1 \text{ mA cm}^{-2}$) to determine the onset potential because the overpotential value is really hard to read from the polarization curve at such small current density.

(4) Line 232 in page 13. “whereas only one broad peak at 6.5 eV(P-Se)...” What the broad peak around 10 eV represents, O?

Response: We thank the reviewer for the careful question. On the basis of your question we referred back to the literatures and found that the peaks in the region from 10 to 11 eV could be assigned to the P 3s band (Synthetic Met., 1991, 45, 203). We thus assign this broad peak around 10 eV to the P 3s band and we modified the related Figure and descriptions in the revised MS.

(5) XAS data (Figures 4c&d, Figure S4): Plot data from Co and Se references together. Which features in Figures 4c&d and Figure s24 represent “electron-deficient Co/Se sites and electron-rich P sites in o-CoSe₂-P” (line 237 in page 13)? Fitting of EXAFS should be done concurrently using Se K edge and Co K edge data, and evaluate coordination numbers (CNs) of Co-Co, Co-Se, Se-Se, and Se-Co, as well as bonding distances, and make sure if CNs of Co-Co and Se-Se show low values (close to zero). For example, see a ref (A. Frenkel, Solving the 3D structure of metal nanoparticles, Z. Kristallogr. 222, 605-611 (2007)).

Response: We thank the reviewer for the insightful comments and questions. Following your suggestion, we have added the Co and Se reference XAS data for comparison, seeing **Figure 4c and d** in the revised MS and **Supplementary Figure 27** in the revised SI. We studied our o-CoSe₂/P catalyst by XAS and EELS. The altered white-line intensities in normalized XANES could be used to speculate the oxidation state of the center atoms (Nat. Commun., 2017, 8, 15938). **Figure 4c** shows the XANES of Se K-edge, where the increased white-line intensity of o-CoSe₂/P as compared to c-CoSe₂ indicates the electron deficiency of Se atoms in o-CoSe₂/P, while Se foil bears oxidation state of 0, thus much strong intensity. **Supplementary Figure 27a** shows a higher shifted edge position of Co K-edge for o-CoSe₂/P sample (also see inset figure), which suggests the loss of electrons in Co atoms. The electron deficiency of Se and Co atoms is further evidenced by L-edge EELS spectra, as shown in **Supplementary Figures 28-29 and their captions**. On the other hand, the shifted edge position and depressed white-line peak of P K-edge reveals the electron-rich P sites (see **Supplementary Figure 30**).

Following your suggestion, we have added the EXAFS fitting results of Se K-edge for both c-CoSe₂ and o-CoSe₂/P, as shown in **Supplementary Figure 27**. The calculated

*parameters were provided in the revised **Supplementary Table 3**. We note that pyrite and marcasite are polymorphs, each Co atom is octahedrally coordinated to six Se atoms and each Se atom is tetrahedrally coordinated to one Se and three Co atoms in both structures. Therefore, the Co-Co bonds are not existent in these structures. And the CNs of Se-Se was reduced due to the partial Se atoms were replaced by P. We further note that, owing to the low P-doping content in the o-CoSe₂/P structure, the Se-P scattering paths could not be well fitted. It is also unlikely to fit the Se-Se scattering path, owing to the similar R values of Se-Co and Se-Se.*

(6) English needs to be refurbished.

Response: *Thanks for the suggestion, the English has been carefully refurbished and now it is much better to read.*

REVIEWERS' COMMENTS:

Reviewer #1 (Remarks to the Author):

Authors have done a good job addressing my concerns. Manuscript can now be accepted.

Reviewer #2 (Remarks to the Author):

Please cite the corresponding references in Line 237 " broad peaks at 6.5 eV (P-Se) and 10.1 eV (P 3s) were detected for o-CoSe₂|P " for the assignments of P-Se and P 3s peaks in Figure 4b.

Reviewer #3 (Remarks to the Author):

I am satisfied with the authors' answers and revisions. I consider that the article is now suitable for publication to Nature Communications.

We thank all the reviewers again for their valuable comments and questions that help us further improve the revised manuscript.

REVIEWERS' COMMENTS:

Reviewer #1 (Remarks to the Author):

Authors have done a good job addressing my concerns. Manuscript can now be accepted.

Response: We thank the reviewer for strong support on the publication of this work.

Reviewer #2 (Remarks to the Author):

Please cite the corresponding references in Line 237 " broad peaks at 6.5 eV (P-Se) and 10.1 eV (P 3s) were detected for *o*-CoSe₂|P " for the assignments of P-Se and P 3s peaks in Figure 4b.

Response: Thanks for the kind suggestion. The suggested reference has been properly cited in the revised Manuscript.

Reviewer #3 (Remarks to the Author):

I am satisfied with the authors' answers and revisions. I consider that the article is now suitable for publication to Nature Communications.

Response: We thank the reviewer for strong support on the publication of this work.